# The Treatment Effect of Liver Transplantation versus Liver Resection for HCC: A Review and Future Perspectives

**DOI:** 10.3390/cancers13153730

**Published:** 2021-07-24

**Authors:** Berend R. Beumer, Roeland F. de Wilde, Herold J. Metselaar, Robert A. de Man, Wojciech G. Polak, Jan N. M. Ijzermans

**Affiliations:** 1Department of Surgery Division of HPB & Transplant Surgery, Erasmus MC Transplant Institute, Erasmus MC, University Medical Centre Rotterdam, 3015AA Rotterdam, The Netherlands; b.beumer@erasmusmc.nl (B.R.B.); r.dewilde@erasmusmc.nl (R.F.d.W.); w.polak@erasmusmc.nl (W.G.P.); 2Department of Gastroenterology and Hepatology, Erasmus MC Transplant Institute, Erasmus MC, University Medical Centre Rotterdam, 3015AA Rotterdam, The Netherlands; h.j.metselaar@erasmusmc.nl (H.J.M.); r.deman@erasmusmc.nl (R.A.d.M.)

**Keywords:** hepatocellular carcinoma, liver resection, liver transplantation, survival, regression discontinuity, Milan criteria

## Abstract

**Simple Summary:**

For patients with early-stage hepatocellular carcinoma, it is important to know whether liver transplantation offers a survival benefit over liver resection. Patients receiving transplantation often have different characteristics in terms of their cancer stage and liver function compared to those being resected. This makes a comparison of the two treatment modalities challenging. This article presents a comprehensive review of research articles comparing these two treatments and discusses their strengths and weaknesses. Furthermore, we suggest a new research design that uses a treatment guideline to help ensure that the groups are more comparable. Hereby, we enable future studies to assess whether liver transplantation offers a survival benefit over liver resection in patients that are eligible for both treatments.

**Abstract:**

For patients presenting with hepatocellular carcinoma within the Milan criteria, either liver resection or liver transplantation can be performed. However, to what extent either of these treatment options is superior in terms of long-term survival is unknown. Obviously, the comparison of these treatments is complicated by several selection processes. In this article, we comprehensively review the current literature with a focus on factors accounting for selection bias. Thus far, studies that did not perform an intention-to-treat analysis conclude that liver transplantation is superior to liver resection for early-stage hepatocellular carcinoma. In contrast, studies performing an intention-to-treat analysis state that survival is comparable between both modalities. Furthermore, all studies demonstrate that disease-free survival is longer after liver transplantation compared to liver resection. With respect to the latter, implications of recurrences for survival are rarely discussed. Heterogeneous treatment effects and logical inconsistencies indicate that studies with a higher level of evidence are needed to determine if liver transplantation offers a survival benefit over liver resection. However, randomised controlled trials, as the golden standard, are believed to be infeasible. Therefore, we suggest an alternative research design from the causal inference literature. The rationale for a regression discontinuity design that exploits the natural experiment created by the widely adopted Milan criteria will be discussed. In this type of study, the analysis is focused on liver transplantation patients just within the Milan criteria and liver resection patients just outside, hereby ensuring equal distribution of confounders.

## 1. Introduction

Over the years, a multitude of treatment options and strategies for hepatocellular carcinoma (HCC) were developed. Of these, liver resection (LR) and liver transplantation (LT) remain the pillars of curative treatment. For the subset of patients with early HCC (i.e., within the Milan Criteria (MC)) that have a good liver function and no portal hypertension, both treatments can be performed [1,2]. This subset is sizable and consists of 20–25% of the surgical HCC population [3,4,5]. Furthermore, the transplant policy for this subset of patients has a wider impact, as the availability of donor organs affects all transplant patients. To what extent either of these two treatments is superior in terms of long-term survival continues to be a topic of debate. Direct comparison of the two treatments is complicated by a myriad of selection processes such as transplantation criteria, waiting list dropout, surveillance schemes, and access to health care. Yet, comparisons of the two treatments are frequently attempted as the difference in 5-year overall survival (OS; i.e., the treatment effect) is the fundament of many discussions. These range from the preferred first-line treatment and prioritisation, the role of salvage transplantation, to the ethicality of living donor liver transplantation (LDLT) [6]. Therefore, this review focuses on the current evidence comparing LT to LR with special attention to lacunas and logical conflicts besides the application of causal inference techniques as the best alternative to a randomised controlled trial (RCT).

## 2. Liver Resection vs. Liver Transplantation: Current Status

There is an abundance of literature comparing LT vs. LR in patients with early-stage HCC. Here we aim to summarise the general trends and differences between different types of studies rather than focusing on a single category. Therefore, we first identified all systematic reviews. Hereafter, we identified all original studies comparing LT to LR published after the last systematic review (i.e., between January 2017 and July 2021) so that all available literature were covered. The articles were identified by performing a search of Embase, Web of Science, the Cochrane database and Google Scholar using variants of the search terms “hepatocellular carcinoma”, “liver transplantation”, and “liver resection” (Appendix A). Additionally, studies were identified through reference lists and by reviewing all articles that cited landmark studies or systematic reviews. The included articles had to describe the survival difference for patients with HCC receiving first-line LR or LT (Figure 1A,B). In the following section, we provide an overview of their conclusions and discuss their strengths and weaknesses. Important to note is that, as these systematic reviews focus on studies that estimate the treatment effect in different ways, we organised the discussion by bundling studies that addressed or were exposed to similar types of biases.

We found that the earliest reviews were subject to the largest biases. These reviews estimated the treatment effect based on single-arm studies that focused on either LT or LR [7,8]. Besides, these reviews included data published between 1985 and 2004, using registries that may not be representative of the treatment modalities nowadays. To remedy the population differences across centres, updated systematic reviews use figures from studies that compare the LT and LR groups within the same cohort [9,10,11,12,13]. Nonetheless, this type of analysis is still at high risk of bias as it does not account for the waiting list dropout. Therefore, results are likely biased in favour of LT with a pooled odds ratio (OR) of the 5-year OS ranging from 0.54 to 0.62 in favour of LT (Table 1). Besides the pooled results from the meta-analyses, all individual studies that were included in these systematic reviews concluded that LT is superior to LR. Original articles published after the latest of these systematic reviews were in favour of LT [14,15,16,17,18,19,20,21,22,23,24,25], except for the study of Eilard et al. (2021), which studied more comparable cohorts limiting the inclusion criteria to patients with Child–Pugh A presenting with HCC within the MC [26]. However, disentangling the relative contribution of liver disease, tumour load and the dropout rate is largely impossible, thus making it challenging to interpret what is exactly being measured.

To account for the dropout of patients on the waiting list, an intention-to-treat (ITT) analysis is preferred. Currently, there are four systematic reviews covering ITT studies (Table 1) [9,10,27,28]. These differed in terms of inclusion criteria with more or less strict requirements in terms of liver cirrhosis and cancer stage. Despite that these studies account for the dropout rate, the pooled OR in this group of studies ranged from 0.6 to 1.19. This heterogeneity in treatment effect is ascribed by the authors to differences in liver function and tumour load between the treatment groups for which the analysis did not adjust [9,27]. Nevertheless, these reviews all conclude that patients treated with LR or LT have a comparable 5-year OS. The two ITT studies published after the last review confirmed these results [29,30].

In addition to comparable survival, it is reported that Disease-Free Survival (DFS) is longer for LT compared to LR with pooled ORs ranging from 0.18 to 0.76 in favour of LT. It is important to note, however, that ITT analysis cannot be performed for DFS, as without curative treatment, patients cannot experience recurrence. Therefore, the long DFS for the LT could partially be explained by the dropout of patients with the most aggressive tumours during the waiting period. An alternative explanation is that LT offers the widest possible resection margin and replaces the diseased remnant liver, hereby also treating the underlying liver disease. This lowers the recurrence rate as the remnant liver can host intrahepatic metastasis and is at higher risk for de novo lesions. The evidence for the latter is provided by studies comparing LR to LDLT patients, who have a substantially shorter wait time compared to deceased donor liver transplant patients. These studies confirm the higher DFS in the LT group [18,31,32,33] (Appendix A). However, caution with generalising results from LDLT to LT is needed because of the high degree of patient selection in the LDLT setting [34,35]. Nevertheless, it is likely that both explanations regarding dropout and the largest possible resection margin are true, although the relative contribution of each is difficult to measure and might differ from study to study. Interestingly, all reviews summarising ITT studies conclude that 5-year OS between LR and LT is comparable. This seems contradictory to the DFS finding. If we accept that, independent of the dropout rate, recurrences occur more frequently in the LR group, this should translate into worse long-term survival. Therefore, indicating that differing DFS and the comparable OS arguments cannot coexist.

As this logical conflict might be caused by differences in baseline characteristics, propensity score matching (PSM) can be used. To the best of our knowledge, there are currently only four studies using PSM in their analysis and no systematic review. Shen et al. (2016) reported for patients presenting within the MC a 5-year OS of 63.6% for LR and 83.2% for LT. The authors concluded, based on the log-rank test, that there was no difference between the two treatments [36]. However, an inspection of the Kaplan–Meier curves revealed that the proportional hazards assumption was violated, weakening this claim. Whereas Shen et al. (2017) reported for patients presenting outside the MC a 5-year OS of for 17.5% LR and 63.2% for LT (*p* = 0.003) [37]. In 2017, Liu et al. published a PSM study, albeit with follow-up too short to report 5-year survival [38]. Lastly, Kaido et al. (2015) reported a 5-year OS of 53% for LR and 63% for LDLT (OR = 0.842 95%CI (0.433–1.638), *p* = 0.613) [33]. They concluded that a survival benefit was not apparent in the LDLT group. An important methodological weakness of these PSM studies is that none of them performed an ITT analysis. Furthermore, it should be noted that PSM is sensitive to the collection of observed variables and does not account for hidden confounders.

Therefore, ITT RCT remains the golden standard for a head-to-head comparison of the two treatments. Only this type of study holds that patients in the intervention group would have attained a similar survival to the control group if they were given the control treatment, hereby giving an unbiased representation of the treatment effect. Nevertheless, the Cochrane review by Taefi et al. (2013) focused on RCTs, concluded that none were performed [39]. Furthermore, no studies are currently registered at clinicaltrials.gov (accessed on 16 July 2021). Despite the necessity of an RCT, they are believed to be infeasible due to the large required sample size and differences in practice patterns [40]. This was recently confirmed by Mazzaferro et al. (2020), who reported on a prematurely terminated RCT that could not be completed according to protocol due to the low accrual and a change in LT policy [41]. In the latter study, ultimately, 43 patients with huge HCC downstaging vs. downstaging + LT were compared. In the group that received only downstaging, 3 patients underwent LR, whereas 21 patients were in the group that underwent LT. Due to the limited sample size and the aggregation over all downstaging techniques, a valid comparison between LT and LR could not be performed.

## 3. Novel Perspectives to Determine Outcome after LT vs. LR

The widely adopted Milan Criteria (MC) provide an opportunity to approximate the treatment effect of LT from large retrospective data registries using causal inference techniques. More specifically, the regression discontinuity introduced by Thistlethwaite and Campbell (1960) may help to find a definitive answer [42]. This research design uses data from patients that are eligible for both treatments and focuses on patients just above and just below a known treatment threshold. The idea (and assumption) is that patients and doctors have limited to no control over the number of tumours and their size, the main determinants for patients to be eligible for LT [1,43]. Therefore, patients transplanted just within the MC are similar, in terms of confounders, to patients who presented just outside the MC and received LR. Therefore, one can argue that patients are randomised based on the time it took them to be diagnosed with HCC. An analysis that exploits this randomisation severely reduces the impact of (unobserved) confounders and approaches the level of evidence from an RCT [44].

The regression discontinuity design in a transplantation context exploits the thresholds of the MC, at which the treatment effect is revealed. The MC state that a patient is eligible for LT if the patient has (1) a single tumour less than 5 cm in diameter or (2) less than three tumours with each less than 3 cm in diameter [45]. First, we confine the discussion to patients with a single lesion HCC that are treated with LT or LR strictly based on the MC. Figure 2 illustrates how the treatment effect *T* can be identified with a regression discontinuity design. Ideally, we could observe the 5-year survival of both treatments for the same patient. In that case, *T* can be calculated by simply subtracting the survival after LR from the survival after LT. Unfortunately, only one of the outcomes can be observed; the outcome that is not observed is called the counterfactual (Figure 2). These values need to be approximated/predicted as then the calculation of *T* is trivial.

If it is assumed that tumour size cannot be controlled and, using appropriate inclusion criteria, it is ensured that all patients can receive both treatments. Then, patients near the 5 cm threshold are similar in terms of (unobserved) confounders because treatment of these patients is randomised by the growth rate of their tumour and the time to diagnosis. In the example of Figure 2, patient *i* was diagnosed just in time, and patient *j* was slightly too late to receive LT. Therefore, a reasonable approximation of *T* can be made by subtracting the outcome of patient *j*, treated with LR, from the outcome of patient *i*, who received LT. Certainly, the analysis of just two patients leads to widely varying estimates if it is repeated in a different sample. Furthermore, the effect of a small difference in tumour size is ignored. Better estimates can be obtained using a regression model that also uses data points slightly farther away from the threshold. The average, conditional on tumour size, is then used to predict the counterfactual. In that case, *T* is the average distance between predicted and observed values.

The setup can be generalised by relaxing the linearity assumption using a more flexible polynomial or spline regression model. Furthermore, at the cost of additional assumptions, regression discontinuity design can be combined with propensity score weighting [46]. However, it remains that the farther from the threshold, the more likely the assumptions regarding the functional form are violated. In order to find the optimal balance between biased vs. unstable estimates, cross-validation can be used in which repeated bandwidths are tried [47].

Furthermore, the analysis can be extended by analysing all thresholds created by the MC (Figure 3). The regression discontinuity design introduced above focused on identifying the magnitude of ridge *d*, which describes the 5 cm threshold for patients with a single lesion. However, we can repeat the analysis for ridges *a* and *b*. Ridge *a* describes patients with tumours smaller than 3 cm where the treatment is based on whether the tumour number exceeds 3. Whereas ridge *b* describes patients with two to three lesions with the treatment based on whether the maximum diameter exceeds 3 cm.

The analysis for threshold *c* of the MC is, however, more complicated. Ridge *c* concerns patients with a maximum tumour diameter between 3 and 5 cm, with the LT group only consisting of patients with single lesions, whereas the LR group only consists of patients with two or more tumours. An important assumption in the regression discontinuity design is that all other factors determining 5-year survival must behave smoothly at the threshold. Otherwise, the ridge is a compound of both disease and treatment factors. The change from single lesion to multinodular disease might not be smooth. Therefore, we must be aware that this assumption might be violated for ridge *c*. If the data size permits, we hypothesise the assumption can be tested at the regions marked with an asterisk.

Similarly to analysing all thresholds created by the MC, regression discontinuity analysis can be performed for other treatment criteria such as the up-to-seven criteria or the Tokyo criteria that lie slightly beyond the MC. Including information from multiple treatment thresholds not only makes more efficient use of the available data but also reveals how the treatment effect changes for different tumour number and size combinations.

However, similar to an RCT, there will be imperfect compliance. Although it is likely that the MC is strongly correlated with receiving LT, the treatment decision depends on a more complex interplay between patient condition, liver function, tumour location and extent of the disease. Tumour characteristics such as size and number might also be modified by means of downstaging, making the threshold fuzzy. Therefore, another extension is to use logistic regression to model the threshold as a smooth change in probability of receiving LT rather than a sharp switch between 0 and 1 [48].

Lastly, it is important to note that although this research design mimics an RCT, careful setup of the analysis is still required. Most importantly, only patients should be included that could have received both treatments. Therefore, studies should exclude patients with decompensated liver disease (Child–Pugh B/C), poor location of the tumour, or portal hypertension from their analysis. Furthermore, similar pitfalls of RCTs should be avoided. For instance, one should be aware that multiple versions of the treatment could exist. One can imagine different versions of LT in terms of pre-treatment (e.g., no pre-treatment, transarterial chemoembolisation (TACE), radio frequency ablation (RFA)) or in terms of transplantation (e.g., emergency, regular transplant, split liver, or LDLT). If ignored, the measured treatment effect will be the weighted average of the treatment effects for each of the versions. At the cost of introducing statistical assumptions, it is possible to control for these factors using traditional statistics. Additionally, marginal/stratified analyses could be performed and used for sensitivity analysis. These could also be reported directly alongside a precise specification of the treatment version [49,50]. To what extent different treatment versions pose a problem is context-dependent. It is likely that ignoring treatment versions no-TACE vs. TACE influences the results more than what type of beads were used during TACE procedures. Furthermore, it is important that the analysis should be conducted based on the ITT as otherwise, dropout can distort the results. Lastly, we want to note that in the last section, the discussion was limited to the difference in 5-year overall survival. However, regression discontinuity analysis would work equally well for other outcomes. In fact, we suggest that researchers also include statistics regarding recurrences and survival after 10-years as this would provide a complete picture. Additionally, we hypothesize that when a longer period is considered, differences between the two treatments become larger, thereby increasing the power of the analysis.

## 4. Conclusions

The survival advantage of LT vs. LR is fundamental in almost all LT discussions. Yet, after three decades of LT and over a hundred papers discussing the comparison with LR, a consensus on the treatment effect is still lacking. Truth finding in this research area is complicated by selection processes. It is likely that subtle differences in, for instance, surveillance schemes, waiting list dropout, and access to health care contribute to the dispute about the size of the treatment effect. In the currently available literature, most studies in which no ITT analysis is performed conclude that LT is superior to LR, whereas studies using an ITT analysis conclude that survival is comparable. Furthermore, these studies state that the DFS is longer in the LT group compared to the LR group. However, none of the ITT studies provide an explanation for these seemingly inconsequential recurrences. We conclude that studies providing a higher level of evidence are needed in an era in which imaging and treatment modalities have improved significantly, and the incidence of early-stage HCC is rising. As no RCTs on this subject will be performed in the near future, we propose a novel research design from the causal inference literature. The regression discontinuity design exploits the natural experiment created by the widely adopted MC. This focuses the analysis on LT patients just within the MC, and LR patients just outside the MC, hereby ensuring equal distribution of (unobserved) confounders similar to an RCT.

## Figures and Tables

**Figure 1 cancers-13-03730-f001:**
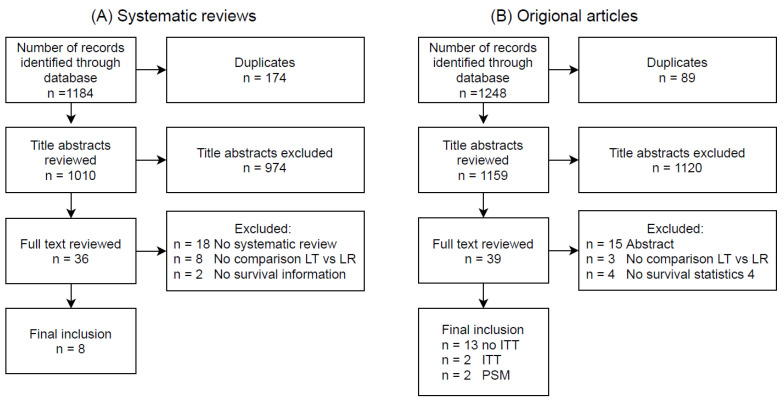
Study selection flowchart. (**A**) systematic studies and (**B**) of original articles published in or after 2017. Abbreviations: number of studies (N); liver transplantation (LT); liver resection (LR); intention-to-treat (ITT); propensity score matching (PSM).

**Figure 2 cancers-13-03730-f002:**
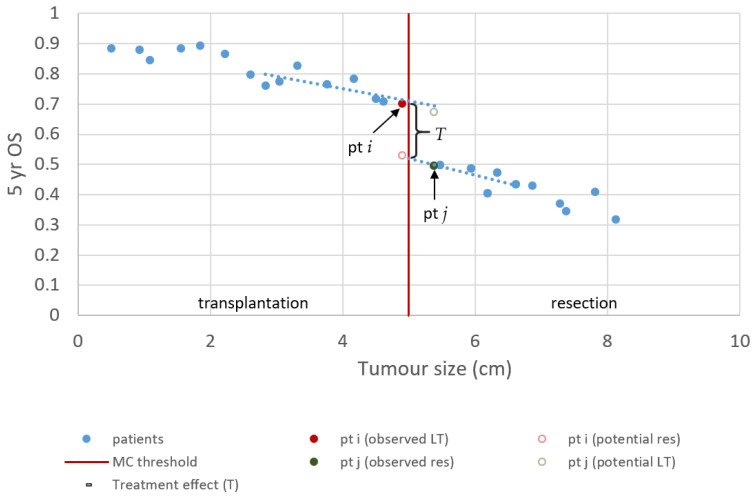
Regression discontinuity for single lesion Milan criteria. The solid points resemble the observed 5-year survival of fictive patients with single lesion HCC. The hollow points resemble long-term survival that could have been observed if the patient was treated with the alternative treatment. In this illustration, we assume that a smaller tumour is preferred over a bigger tumour, and that this relationship is linear. Furthermore, we chose to depict LT as the superior treatment, with a gain in survival T.

**Figure 3 cancers-13-03730-f003:**
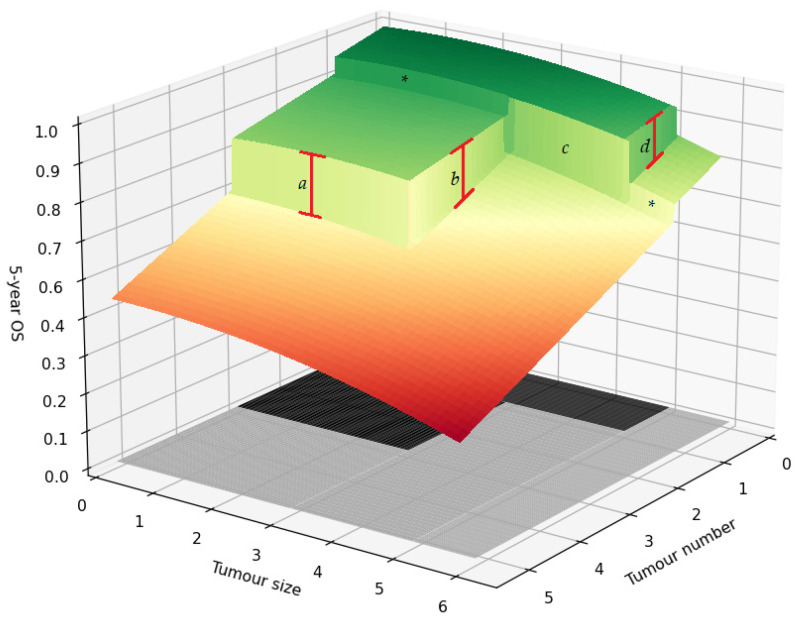
Regression discontinuity for the Milan criteria. Illustration of the average 5-year survival of patients for different combinations of tumour number and size. The black area on the bottom describes the region in which patients are within the Milan criteria and are treated with liver transplantation. The area in grey marks the region outside the Milan criteria where patients are treated with liver resection. Lines *a*, *b*, *d* depict the treatment effect of liver transplantation vs. liver resection. The boundary marked by *c* describeds the comparison between LT and LR in which the LT group only consists of patients with single lesions, whereas the LR group only consists of patients with two or more tumours. Lastly, the asterisks indicate the regions where the smoothness assumption can be tested.

**Table 1 cancers-13-03730-t001:** Meta-analyses of systematic reviews.

Author	Year of Publication	Population	5-Year OS					
			Number of Studies	Number of Patients	Pooled OR (95%CI)	*p*-Value	I^2^	Conclusion
Non Intention-To-Treat								
Rahman [10]	2012	Non-ITT	8	1769	0.62 (0.50–0.76)	<0.001	0	In favour of LT
Dhir [9]	2012	Within MC	10	1763	0.58 (0.36–0.94)	0.027	78	In favour of LT
Dhir [9]	2012	Within MC. Well compensated cirrhosis	6	994	0.54 (0.38–0.77)	0.001	32	In favour of LT
Schoenberg [11]	2017	Within MC. No Child-Pugh class C	8	1221	0.60 (0.45–0.78)	<0.001	70	In favour of LT
Xu [12]	2014		15	4102	0.54 (0.44–0.67) *	-	40	In favour of LT
Zheng [13]	2014		45	8288	0.56 (0.46–0.69) *	<0.001	65	In favour of LT
Intention-to-treat								
Rahman [10]	2012	ITT	4	1310	1.19 (0.78–1.80)	0.042	65	In favour of LR
Dhir [9]	2012	ITT. Within MC	6	1118	0.60 (0.29–1.24)	0.166	83	In favour of LT
Dhir [9]	2012	ITT. Within MC. Well compensated cirrhosis	3	412	0.52 (0.30–0.91)	0.022	31	In favour of LT
Proneth [14]	2014	ITT	7	1572	0.84 (0.48–1.48)	0.055	84	In favour of LT
Menahem [15]	2017	ITT	9	1428	0.60 (0.32–1.02)	0.060	80	In favour of LT

Overview of the meta-analyses from systematic reviews for overall survival (OS). All systematic reviews included studies that described patients with hepatocellular carcinoma receiving liver resection or liver transplantation as first-line therapy. Additional criteria defining the study population are listed in the table. Odds were calculated as the number of deaths divided by the number of patients alive. Odds ratio (OR) was calculated as liver transplantation odds divided by liver resection odds. Note: * To set the reference group to liver resection, the odds ratio reported in the systematic review was inverted. Abbreviations: patients (pt). Child-Pugh class (CP).Intention-to-treat (ITT). Milan Criteria (MC).

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
