# Peer review of "The Treatment Effect of Liver Transplantation versus Liver Resection for HCC: A Review and Future Perspectives"

_cancers, 2021, doi:10.3390/cancers13153730_

Round 1

Reviewer 1 Report

Beumer et al. reported their review and future perspectives related to the treatment effect of liver transplantation versus liver resection for HCC. The authors used the regression discontinuity design exploits the natural experiment created by the widely adopted Milan Criteria (MC). This focuses the analysis on LT patients just within the MC, and LR patients just outside the MC, trying to ensuring equal distribution of (unobserved) confounders similar to an RCT. There are some minor concerns raised.

  1. The discussion about the LT and LR in the OS and DFS in the study is critical. The major importance in the comparison of both treatments will be in the similar patients’ conditions in the baseline characteristics. Is it possible to also explore the regression discontinuity for a < 3 lesions group of patients within MC as the single lesion patient group in Figure 1?
  2. In figure 2, the gain in the survival T seems obvious. Can authors try to elucidate the T in 1, 2, 3, 4 cm of HCC by regression discontinuity design?
  3. For figure 2 showing regression discontinuity for average 5-year overall survival for MC. How about the 5-year DFS with the limited data?
  4. Authors are encouraged to provide a flowchart of study inclusion for Table 1.

Author Response

Point 1: The discussion about the LT and LR in the OS and DFS in the study is critical. The major importance in the comparison of both treatments will be in the similar patients’ conditions in the baseline characteristics. Is it possible to also explore the regression discontinuity for a < 3 lesions group of patients within MC as the single lesion patient group in Figure 1?

Response 1: We thank the reviewer for this remark. Certainly, it is possible to explore the treatment effect for patients with less than 3 HCC lesions. For these patients there is a threshold that can be used, similar to the one for a single lesion as depicted in Figure 1. Figure 2 shows this in the 3D generalization at ridge b which contrasts patients with 2-3 lesions and maximum diameter <3 cm receiving LT vs. patients with 2-3 lesions and a maximum tumour diameter of > 3cm receiving LR. We have clarified this specific point in lines 243- 246 in the manuscript as follows:

“Ridge a describes patients with tumours smaller than 3 cm where the treatment is based on whether the tumour number exceeds 3. Whereas ridge b describes patients with 2 to 3 lesions with the treatment based on whether the maximum diameter exceeds 3 cm.”

Point 2: In figure 2, the gain in the survival T seems obvious. Can authors try to elucidate the T in 1, 2, 3, 4 cm of HCC by regression discontinuity design?

Response 2: In Figure 2 is indeed depicted that the treatment effect is revealed at the “borders” of the Milan criteria. As the borders of the Milan criteria are fixed, we cannot observe the treatment effect elsewhere using regression discontinuity alone. Matching in conjunction with RD can provide estimates in these regions, although this requires more statistical assumptions. Alternatively, data of patients treated under a different set of criteria could be used. For example, with the up-to-seven criteria or Tokyo criteria the boundaries lie slightly further outward. Combining multiple regression discontinuity analyses in this manner would result into a clearer picture of how the treatment effect changes among patients. We have clarified this point in lines 234-237 and 265-269 as outlined below:

“Furthermore, at the cost of additional assumptions, regression discontinuity design can be combined with propensity score weighting [31]. However, it remains that the farther from the threshold the more likely the assumptions regarding the functional form are violated.”

And: “Similarly to analysing all thresholds created by the MC, regression discontinuity analysis can be performed for other treatment criteria such as the up-to-seven criteria or Tokyo criteria that lie slightly beyond the MC. Including information from multiple treatment thresholds not only makes more efficient use of the available data, but it also reveals how the treatment effect changes for different tumour number and size combinations.”

Point 3: For figure 2 showing regression discontinuity for average 5-year overall survival for MC. How about the 5-year DFS with the limited data?

Response 3: Figure 2 is an illustration for OS, but the technique would work equally well for DFS if data is available. We have added this suggestion in the concluding paragraph of section 2 in lines 296 to 300 as follows:

“Lastly, we want to note that in the last section the discussion was limited to the difference in 5-year overall survival. However, regression discontinuity analysis would work equally well for other outcomes. In fact, we suggest that researchers also include statistics regarding recurrences and survival after 10-years as this would provide a more complete picture.”

Point 4: Authors are encouraged to provide a flowchart of study inclusion for Table 1.

Response 4: A flowchart depicting the in- and exclusion criteria of assessed studies is illustrated in Figure 1A.

Reviewer 2 Report

In their article entitled "The Treatment Effect of Liver Transplantation versus Liver Resection for HCC: A Review and Future Perspectives", Beumer and colleagues explore the different approaches to the analysis of transplantation and resection for early-stage HCC.

This paper basically represent a mini-review on the available systematic reviews/meta-analyses an ITT studies published so far: the Authors conclude that based on the current evidence, we can not draw definitive conclusion on what treatment (LT or LR) is the most effective, mainly because of the lacking or RCT.

In order to overcome the objective difficulty to perform RCTs, the Authors offers an original insight with the application of regression discontinuity design, which is extensively explained in the text.

I suggest the Authors to discuss the problem of patient prioritization for LT on waiting list in the introduction section, referring to the following research 10.3390/cancers11060741

Best regards

Author Response

Point 1: I suggest the Authors to discuss the problem of patient prioritization for LT on waiting list in the introduction section, referring to the following research 10.3390/cancers11060741

Response 1: We are thankful to the reviewer for this suggestion. Prioritization is certainly a subject in which the treatment effect is key. Therefore, we have added this matter in the introduction in lines 53-56.

“Yet, comparison of the two treatments is frequently attempted as the difference in 5-year overall survival (OS) (i.e., the treatment effect) is the fundament of many discussions. These range from the preferred first-line treatment and prioritization, the role of salvage transplantation, to the ethicality of living donor liver transplantation (LDLT) [6].

Reviewer 3 Report

This paper summarized the findings from previous review papers that comparing the treatment effects and survival of liver transplantation and liver resection and proposed a novel approach using causal inference theory to evaluate these data in future research. This is an important research question and the current manuscript correctly pointed out the previous studies did not fully address the issue of intention-to-treat and variation in disease severity among patient received both treatments. The proposed method which stems from the counterfactual theory is feasible and potentially innovative. However, some notable shortcomings should be addressed to improve the manuscript, which is detailed below.

Major comments:

  1. The authors only summarized results from review papers but not original research, which is a major limitation of the current review. There is no update on the research conducted in the field in the last 5 years (latest review paper published in 2017). As a review paper, what more does the current review offer beyond the previous review? The current review could be substantially strengthened by reviewing results from recent major studies and provide an update for the field.
  2. There is no mention of the effect of TACE on LT vs. LR treatment outcomes. TACE has been administered to HCC patients to fit into the Milan Criteria. Whether it may impact the overall survival for patients who received TACE LT vs. no-TACE LT vs. LR patients is unclear.   
  3. While I appreciate the authors' proposed methods take into account counterfactual theory to control for patient variability and ITT in their proposed method, certain details will still need to be considered in the proposed method. For example, as mentioned above, TACE treatment before LT and LR, type of liver transplantation, etc. could play a major part in the outcomes and survival of the specific treatment. It is unclear how they will capture these variations in their study design.

Minor comments:

  1. Table 1 is confusing and will need major restructuring. It is unclear what Odds ratio is reported, who is the reference group, what population they are, how they were evaluated, what type of review (systemic? Meta? Other?) and why these groups were combined or separated the way it is.

Author Response

Major comments:

Point 1: The authors only summarized results from review papers but not original research, which is a major limitation of the current review. There is no update on the research conducted in the field in the last 5 years (latest review paper published in 2017). As a review paper, what more does the current review offer beyond the previous review? The current review could be substantially strengthened by reviewing results from recent major studies and provide an update for the field.

Response1: We thank this reviewer for the made suggestions. In order to strengthen our review, we have conducted an additional literature search of the last five years to ensure full coverage of the available literature. The search query and corresponding flowchart are added in Appendix A and Figure 1B, respectively. Selected citations and the most relevant outcomes are included in lines 104-107, 126, and 164-166.  

In summary, compared to the systematic reviews already available, our current study is intended to offer a wider overview of the literature. For example, the review of Menahem et al. focuses on intention-to-treat studies, but leaves comparison with non-ITT studies, matching studies or studies comparing LR vs LDLT  out of scope. As such, we feel that the current review provides a new perspective and forms a solid basis to the proposed new study design.

Point 2: There is no mention of the effect of TACE on LT vs. LR treatment outcomes. TACE has been administered to HCC patients to fit into the Milan Criteria. Whether it may impact the overall survival for patients who received TACE LT vs. no-TACE LT vs. LR patients is unclear.   

Response 2: This matter raised by the reviewer is indeed quite relevant. In the manuscript we have clarified that downstaging leads to a so-called fuzzy threshold. This effect could be accounted for by using a logistic regression to model the probability of receiving LT rather than just assuming a sharp switch between zero and one. The following passage is added in lines 274-277:

“Tumour characteristics such as size and number might also be modified by means of downstaging making the threshold fuzzy. Therefore, another extension is to use a logistic regression to model the threshold as a smooth change in probability of receiving LT rather than a sharp switch between 0 and 1 [49].”

Point 3: While I appreciate the authors' proposed methods take into account counterfactual theory to control for patient variability and ITT in their proposed method, certain details will still need to be considered in the proposed method. For example, as mentioned above, TACE treatment before LT and LR, type of liver transplantation, etc. could play a major part in the outcomes and survival of the specific treatment. It is unclear how they will capture these variations in their study design.

Response 3: Liver transplantation has indeed multiple versions with their own treatment effects. If an overall analysis is performed, assuming TACE impacts survival, then the measured treatment effect will depend on the proportion of TACE vs no-TACE in the transplant population. In order to adress this matter we included the following in lines 283-294.

“Furthermore, similar pitfalls of RCT’s should be avoided. For instance, one should be aware that multiple versions of the treatment could exist. One can imagine different versions of LT in terms of pre-treatment (e.g., no pre-treatment, transarterial chemoembolization (TACE), or radio frequency ablation (RFA)) or in terms of transplantation (e.g., emergency, regular transplant, split liver, or LDLT). If ignored, the measured treatment effect will be the weighted average of the treatment effects for each of the versions. At the cost of introducing statistical assumptions, it is possible to control for these factors using traditional statistics. Additionally, marginal/stratified analyses could be performed and used for sensitivity analysis. These could also be reported directly alongside a precise specification of the treatment version [50,51]. To what extent different treatment versions pose a problem is context dependent. It is likely that ignoring treatment versions no-TACE versus TACE influences the results more than what type of beads were used during TACE procedures.”

Minor comments:

Point 4: Table 1 is confusing and will need major restructuring. It is unclear what Odds ratio is reported, who is the reference group, what population they are, how they were evaluated, what type of review (systemic? Meta? Other?) and why these groups were combined or separated the way it is.

Response 4: We believe that part of the confusion addressed by the reviewer was caused by the incorrectly displayed landscape format. As such, we have focused on OS and relegated information regarding DFS to Appendix B. Furthermore, population and grouping are defined more clearly. Also, the footnote is clarified. This now includes an explicit statement that the data are extracted from meta-analyses performed in systematic reviews and describes what the odds ratios are referring to:

“Footnote Table 1: Overview of the meta-analyses from systematic reviews. All systematic reviews included studies that described patients with hepatocellular carcinoma receiving liver resection or liver transplantation as first line therapy. Additional criteria defining the study population are listed in the table. Odds were calculated as the number of deaths divided by number of patients alive. Odds ratio (OR) was calculated as liver transplantation odds divided by liver resection odds. Note: * To set the reference group to liver resection the odds ratio reported in the systematic review was inverted. Abbreviations: Child-Pugh class (CP), overall survival (OS), Intention-to-treat (ITT), Milan Criteria (MC).”

Round 2

Reviewer 3 Report

I appreciate the authors provided some major revisions to the paper and it has been improved substantially.